# Developing a Natural Language Processing tool to identify perinatal self-harm in electronic healthcare records

Karyn Ayre[1,2]*, André Bittar[3], Joyce Kam[4], Somain Verma[4], Louise M. Howard[1,2], Rina Dutta[2,3]

**1** Section of Women's Mental Health, Health Service and Population Research Department, Institute of Psychiatry, Psychology and Neuroscience, Kings College London, London, United Kingdom, **2** South London and Maudsley NHS Foundation Trust, Bethlem Royal Hospital, Kent, London, United Kingdom, **3** Academic Department of Psychological Medicine, Institute of Psychiatry, Psychology and Neuroscience, Kings College London, London, United Kingdom, **4** King's College London GKT School of Medical Education, London, United Kingdom

☯ These authors contributed equally to this work.

* karyn.k.ayre@kcl.ac.uk

**Data Availability Statement:** Data are owned by a third party, Maudsley Biomedical Research Centre (BRC) Clinical Records Interactive Search (CRIS) tool, which provides access to anonymised data

# Abstract

## Background

Self-harm occurring within pregnancy and the postnatal year ("perinatal self-harm") is a clinically important yet under-researched topic. Current research likely under-estimates prevalence due to methodological limitations. Electronic healthcare records (EHRs) provide a source of clinically rich data on perinatal self-harm.

## Aims

(1) To create a Natural Language Processing (NLP) tool that can, with acceptable precision and recall, identify mentions of acts of perinatal self-harm within EHRs. (2) To use this tool to identify service-users who have self-harmed perinatally, based on their EHRs.

## Methods

We used the Clinical Record Interactive Search system to extract de-identified EHRs of secondary mental healthcare service-users at South London and Maudsley NHS Foundation Trust. We developed a tool that applied several layers of linguistic processing based on the spaCy NLP library for Python. We evaluated mention-level performance in the following domains: span, status, temporality and polarity. Evaluation was done against a manually coded reference standard. Mention-level performance was reported as precision, recall, F-score and Cohen's kappa for each domain. Performance was also assessed at 'service-user' level and explored whether a heuristic rule improved this. We report per-class statistics for service-user performance, as well as likelihood ratios and post-test probabilities.

## Results

Mention-level performance: micro-averaged F-score, precision and recall for span, polarity and temporality >0.8. Kappa for status 0.68, temporality 0.62, polarity 0.91. Service-user

derived from SLaM electronic medical records. These data can only be accessed by permitted individuals from within a secure firewall (i.e. the data cannot be sent elsewhere), in the same manner as the authors. For more information please contact: cris.administrator@slam.nhs.uk.

**Funding:** KA is funded by a National Institute for Health Research Doctoral Research Fellowship (NIHR-DRF-2016-09-042). The views expressed are those of the authors and not necessarily those of the NHS, the NIHR or the Department of Health and Social Care. https://www.nihr.ac.uk/ RD is funded by a Clinician Scientist Fellowship (research project e-HOST-IT) from the Health Foundation in partnership with the Academy of Medical Sciences which also party funds AB. https://health.org.uk/ https://acmedsci.ac.uk/ AB's work was also part supported by Health Data Research UK, an initiative funded by UK Research and Innovation, Department of Health and Social Care (England) and the devolved administrations, and leading medical research charities, as well as the Maudsley Charity. https://www.ukri.org/ https://maudsleycharity.org/ Professor Louise M Howard receives salary support from NIHR South London and Maudsley/ King's College London Biomedical Research Council and the NIHR South London Applied Research Collaboration. https://www.nihr.ac.uk/ The funders had no role in study design, data collection and analysis, decision to publish, or preparation of the manuscript.

**Competing interests:** The authors have declared that no competing interests exist.

level performance with heuristic: F-score, precision, recall of minority class 0.69, macro-averaged F-score 0.81, positive LR 9.4 (4.8–19), post-test probability 69.0% (53–82%). Considering the task difficulty, the tool performs well, although temporality was the attribute with the lowest level of annotator agreement.

## Conclusions

It is feasible to develop an NLP tool that identifies, with acceptable validity, mentions of perinatal self-harm within EHRs, although with limitations regarding temporality. Using a heuristic rule, it can also function at a service-user-level.

## Introduction

Self-harm is defined by the National Institute for Health and Care Excellence as an "act of self-poisoning or self-injury carried out by a person, irrespective of their motivation" [1]. Data from several high-income countries indicates that self-harm is increasingly common, particularly in young women [2, 3]. During pregnancy and the postnatal year, a time known as "the perinatal period", around 5–14% of women are estimated to experience thoughts of self-harm [4]. Yet there remains an evidence gap around acts of perinatal self-harm [5].

Given self-harm is strongly associated with mental disorder [6], this is likely to be the case for perinatal self-harm. It may therefore be a marker of unmet treatment need. Suicide is a leading cause of maternal death and such suicides are frequently preceded by acts of perinatal self-harm [7, 8].

Current evidence regarding the prevalence of perinatal self-harm is mainly derived from studies using administrative hospital discharge datasets which may under-represent the true prevalence [5]. Evidence suggests perinatal self-harm is more common in women with serious mental illness (SMI) [5], meaning this population should be a focus of research.

The widespread use of electronic healthcare records (EHRs) means that large amounts of nuanced clinical information can be centrally stored for large cohorts of service-users. However, free-text documentation means many clinical variables are not readily extractable. A free-text search strategy could identify self-harm synonyms but would lack the contextual "awareness" required to distinguish relevant from non-relevant mentions, such as thoughts of or statements of negation of self-harm.

Natural Language Processing (NLP) can recognise relevant linguistic context (e.g. lexical variation, grammatical structure, negation) and is increasingly used in clinical research to extract information from EHRs [9, 10]. The use of NLP to investigate suicidality is relatively new and the literature is small [11]. However, NLP has been used to identify suicidality in EHRs [12–14], including those of adolescents with autism spectrum disorders [15]; general hospital [16] and primary care attenders [17].

To our knowledge, only one other group has used NLP to identify perinatal self-harm. Self-harm was identified as part of a composite measure of both thoughts of suicide and acts of self-harm [18, 19] and not specifically among women with SMI.

In this study, we aimed to develop an NLP tool for the purpose of identifying acts of perinatal self-harm at a mention and service-user level, within de-identified EHRs of women with SMI.

## Materials and methods

### Data sources

The South London and Maudsley (SLaM) National Institute For Health Research (NIHR) Biomedical Research Centre (BRC) Clinical Record Interactive Search (CRIS) system [20] provides regulated access to a database of de-identified EHRs of all service-users accessing South London and Maudsley NHS Foundation Trust (SLaM), which is the largest secondary mental healthcare provider in the United Kingdom. In this context, "EHR" refers to a single clinical document, within one universal electronic healthcare recording system called the "Electronic Patient Journey System".

CRIS is linked with Hospital Episode Statistics (HES) [21], a database of anonymised clinical, administrative and demographic details of NHS hospital admissions of service-users over the age of 18. By searching for codes indicating delivery, linkage with CRIS has been demonstrated to be a valid way of generating a cohort of women accessing secondary mental healthcare during the perinatal period [22].

### Ethical approval

CRIS has pre-existing ethical approval via the Oxfordshire Research Ethics Committee C (ref 18/SC/0372). Linkage with HES data is managed by the Clinical Data Linkage Service. The BRC has ethical and Section 251 approval to enable linkage with HES (Ref: ECC 3-04(f)/2011). This project was done under the CRIS Oversight Committee approval 16–069 that relates to KA's Fellowship. The CRIS Oversight Committee is chaired by a service-user and member of the SLAM BRC Stakeholder Participation theme.

### Development of coding rules

Self-harm is a complex concept and may be defined in different ways. The clinical validity of describing self-harm based on suicidal intent (e.g. "suicide attempts" versus "non-suicidal self-injury") has been questioned [23]. The NICE definition of self-harm does not incorporate intent [1]. Therefore, when creating a list of synonyms or "keywords" for self-harm, we conceptualised self-harm broadly and utilised several sources: the secondary mental healthcare clinical expertise of the first author, the general literature on self-harm [24, 25] and terms used in other studies of self-harm in EHRs [22, 26–28]. See S1 File for a full list of keywords.

Mentions of these keywords within the EHRs were appropriately annotated in a sample of 131 EHRs pre-selected from previous research into self-harm in pregnant women with affective and non-affective psychotic disorders by Taylor et al [22, 28]. We devised rules regarding the *span* of text to annotate as a mention and how to annotate mention attributes (see S2 File).

**Span.**   Only the keyword within the mention, not the surrounding text or whole sentence, was annotated. The keyword was usually a noun and direct synonym of self-harm, e.g. overdose. Occasionally, the keyword was a noun, but not a direct synonym, e.g. in the phrase "*she had scratches on her arm*" only the indicative noun (i.e. "*scratches*") was annotated. If the keyword was an adjective that modified a noun, it was annotated along with the noun it described, e.g. in the phrase "*she had a self-harming impulse*", both "self-harming" and "impulse" were annotated. Where the keyword was a verb, the direct object noun/pronoun that it related to was also annotated, e.g. in the phrase "she *cut herself*", both "cut" and "herself" were annotated. Occasionally, the verb implied a passive or non-deliberate action. For example: "*she climbed out a window and fell off*". Falling is a passive or unintentional event, as opposed to jumping. However, in this case the prior act of climbing indicates an active element. Although falling is

passive, it was the fall that caused harm. Therefore, the verb, the pronoun it related to and the intervening words "*fell off*" were annotated.

**Attributes and coding rules.**   We identified three main attributes of mentions of self-harm: status, temporality and polarity. Status specified whether a self-harm event occurred or not. For example, if a mention described thoughts of self-harm, rather than an act of self-harm, they were inferred and annotated to be non-relevant. Mentions of third-party self-harm (e.g. "*her mother took an overdose*") were annotated as non-relevant. We included an "uncertain" category as in a very small number of cases it was not possible, even with whole document context, to determine whether a mention was referring to an act of self-harm or not.

Temporality specified whether an act was current or historical. We were interested in self-harm occurring during pregnancy and/or the postpartum year. As only EHRs created within the service-user's perinatal period were being annotated, non-perinatal temporality was sometimes obvious e.g. "*took an overdose ten years ago*". Events which occurred within one month prior to the EHR were coded as current. This time frame is the same as that used in previous work investigating the prevalence of self-harm in the EHRs of a cohort of pregnant women in CRIS [28] and reflects the standard time period often used in clinical interviews that ask about self-harm, such as the Mini-International Neuropsychiatric Interview [29]. Ambiguous references to chronic events were problematic e.g. "*chronic history of self-harm*". Although this mention describes a chronic occurrence i.e. happening in the past, it also references the fact that the events are potentially ongoing. We decided to code such mentions as current. We initially included an "uncertain" category in order to flag complex cases during manual annotation, although not as an attribute option for the final tool.

Polarity specified whether or not the mention expressed a negation of self-harm (e.g. "*she denied self-harm*"). The purpose of this attribute was to allow the algorithm to filter out negations. Occasionally negation was written using symbols e.g. "*Suicide attempts*: *X*". Here, the meaning of the mention was annotated i.e. polarity negative.

## Manual annotation of a reference standard

For the purposes of developing and evaluating the tool's performance, we created a reference standard, manually annotated, corpus of EHRs. First, we randomly sampled 400 EHRs from Taylor's study of self-harm in pregnant SLaM service-users with affective and non-affective psychotic disorders [22, 28]. All EHRs were independently double-annotated by three annotators (KA, JK, SV) according to the coding rules, using Extensible Human Oracle Suite of Tools (eHOST) software [30]. We measured pairwise inter-annotator agreement in terms of precision (positive predictive value), recall (sensitivity) and F-score (harmonic mean of precision and recall), as well as kappa [31] (agreement adjusted for chance) for attributes. Agreement scores were calculated using the scikit-learn (version 0.21.3) machine learning library for Python [32]. The final reference standard was created by adjudication of disagreements by KA. This was split into development (N = 320 EHRs, 152 service-users) and test (N = 80 EHRs, 59 service-users) sets.

## NLP development

**System description.**   We developed a rule-based tool around spaCy (version 2.1.3), an NLP library for Python. Code for the tool is available online [33]. The tool takes a text as input and applies five processing layers in sequential order, outputting an XML file in which all detected self-harm mentions and their attributes are annotated with XML tags. Each layer of processing adds annotations that are available in subsequent layers. The five processing layers are as follows:

**1. Linguistic pre-processing.** Sentence detection, tokenisation (segmentation of the text into word tokens), part-of-speech tagging (determining the grammatical category of words), lemmatisation (finding the "root" form of inflected words) and dependency parsing (determining the grammatical relations between words). The tokenisation step includes a set of custom tokenisation rules to deal with errors made by spaCy's default tokeniser (e.g. *self-harm*, *self-injury*, *fh/o* which are incorrectly split into several word tokens). The dependency parsing step identifies syntactic relations such as *subject*, *direct object*, *modifier* and *negation*. Dependency parsing has been used in prior work on the analysis of clinical texts, for such tasks as relation extraction [34, 35], identifying family history [36] and negation detection [37].

**2. Lexical rules.** This step consists of tagging of words with a given semantic category according to a set of 13 manually created lexicons. These lexicons include terms for self-harm, body parts, as well as relevant negation and temporal markers. A full list of these lexicons and example content is shown in Table 1.

**3. Token sequence rules.** The final layer of processing consists of a sequence of regular token-based grammars that take into account the context in which words appear. Grammar rules have access to all linguistic features added during pre-processing, as well as semantic categories added during lexical tagging. These rules are applied to detect self-harm expressions in context and correct and update the annotations added by previous processing layers. These rules are used both to detect or exclude mention spans and assign attribute values. A specific set of token sequence rules is used to identify history sections in EHRs. The rule attribute 'name' indicates the unique rule name for development purposes, 'pattern' is the token sequence pattern to match in the text, 'annotation' is the attribute and value that is marked on the recognised token sequence.

**4. Negation detection.** Negation is detected using the syntactic dependency tree for each sentence. Any mention that heads a 'neg' grammatical dependency is annotated as negative (e.g. "*she did not cut herself*"). If a mention's governor is a negated reported speech verb (R_SPEECH), the mention is also assigned negative polarity (e.g. "*she did not report harming herself*"). Finally, any mention governed by a word annotated as NEGATION is also annotated as having negative polarity (e.g. "*she denies any self-harm*").

**5. Contextual search.** To further assign values to attributes for identified mentions, a contextual search is used to detect markers of temporality and status. A window of ten tokens to the left and right of a mention is used as context. If a token labelled 'past' is found within this

**Table 1. Lexicons used for tagging of semantic categories.**

| Category | Example terms | Annotation | Example |
|---|---|---|---|
| Self-harm | *DSH, overdose* | SH | *She took an overdose* |
| Body part | *wrist, hand, torso* | BODY_PART | *She had cut her left wrist* |
| Harm action | *cut, burn, hit, lacerate* | HARM_ACTION | *She lacerated her arm* |
| Family members | *mother, father, daughter* | FAMILY | *Her mother took an overdose* |
| Uncertainty | *plan, prone, risk, thought* | HEDGING | *She would cut herself* |
| Intention | *aim, deliberately, intend* | INTENT | *She cut herself deliberately* |
| Medication | *olanzapine, paracetamol, aspirin* | MED | *She took 12 paracetamol tablets* |
| Modality | *could, would, possible* | MODALITY | *Possibility of self-harm* |
| Negation | *not, never, no, deny* | NEGATION | *Denies self-harm* |
| Reported speech | *say, claim, disclose* | R_SPEECH | *She disclosed having thoughts of cutting herself* |
| Life stages | *adolescent, teenager, young, kid* | LIFE_STAGE | *She started self-harming in her teens* |
| Past references | *previous, past, historical* | PAST | *Previous episodes of self-harm* |
| Present references | *Monday, current, recent* | PRESENT | *Current episode of self-harm* |

window, the mention is labelled as historical. Similarly, if a token labelled 'hedging' or 'modality' is found within the window, the mention is annotated as non-relevant.

### Unusual linguistic cases

During development of the coding rules, we identified unusual examples that did not fit with our pre-defined strategy. This led to the refinement of the tool's processing layers. Some examples are detailed in Fig 1, with the relevant keyword highlighted in italics.

### Further development: Service-user selection heuristic

In our reference standard, the majority of service-users who had self-harmed perinatally had more than one mention in their EHRs (see S1 Table). Based on this, we explored the use of a service-user selection heuristic, whereby we restricted flagging of service users as true positive cases to only those who had two or more mentions of perinatal self-harm in their EHRs.

## Results

### Inter-annotator agreement

Table 2 presents micro-averaged pairwise inter-annotator agreement on mention spans and attributes using precision, recall and F-score and Cohen's kappa (39), within the development set of EHRs (N = 320 documents). Due to the very small number of cases of "uncertain" status and temporality, the high degree of class imbalance means macro-averaged figures were not a fair representation of performance (see S2 Table). All figures are rounded up to 2 decimal points.

**Example 1: "currently well but has a tendency to self-harm when depressed".**

The status of "tendency" was initially non-relevant, as it was a "tendency" rather than an act. However, we felt that if someone is described as having a historical tendency towards doing something, this indicates that the issue in question must have been carried out in the past at least once. The classification of the status of mentions of "tendency" was therefore changed to relevant.

**Example 2: "she reports a planned overdose in the context of suicidal intent at the age of…".**

We debated whether "planned" refers to a plan (which would be status non-relevant) or a "planned overdose" i.e. a physical act of self-harm which was planned. From the first author's clinical experience, the phrase "planned overdose" is commonly used, as pre-planning of an act of self-harm suggests additional clinical risk. The tool structure was changed so that when the adjectival modifier "planned" is used with "overdose" the tool annotates the mention as relevant.

**Example 3: "It appears that these suicidal thoughts and act (when she was aged…)…".**

The keyword in this sentence ("suicidal thoughts") is part of a coordinated noun phrase in which the second conjunct ("act") refers to self-harm. We decided that the status of the entire expression "suicidal thoughts and act" should be annotated as relevant as here "suicidal" is an adjective and it is describing two things – "thoughts and act" together. Thoughts on their own are non-relevant, but here the thoughts are fused with the act. We updated the rules to consider "suicidal thoughts and act" as one expression and to classify its status as relevant.

**Fig 1. Examples of unusual linguistic cases.**

**Table 2. Micro-averaged pairwise inter-annotator agreement.**

|  | Precision | Recall | F-score | Kappa |
|---|---|---|---|---|
| **Span** | 0.83 | 0.89 | 0.85 | N/A |
| **Polarity** | 0.96 | 0.96 | 0.96 | 0.92 |
| **Temporality** | 0.90 | 0.90 | 0.90 | 0.78 |
| **Status** | 0.94 | 0.94 | 0.94 | 0.88 |

## Evaluation of the tool

We evaluated the tool on two levels: mention and service-user. Table 3 shows the micro-averaged mention-level evaluation statistics from both the development (N = 320 documents) and test (N = 80 documents) datasets. Again, class imbalance for status meant macro-averaging was not appropriate (only 9 mentions of "uncertain" status in reference standard, see S3 Table for macro-averaged results) so micro-averaged results are presented.

Service-user-level performance indicates how well the tool identifies service-users who have at least one recorded "true" self-harm mention in <u>any</u> of their EHRs. A "true" mention has the attribute values status = relevant, polarity = positive, temporality = current. We present results with and without the heuristic rule of at least two positive mentions, derived from both the test set (Table 4). When the tool was run with the heuristic, there were no false positives, meaning there were issues with perfect prediction. Total absence of false positives is unlikely to occur in a very large sample and, in this case, most likely indicates the sample size of the test set (N = 59 service-users) is too small for patient-level analysis. We therefore present service-user results in the development set (N = 152 service-users, Table 5).

Due to class imbalance, we report per-class precision, recall and F-score (e.g. precision$_{MAJ}$, precision$_{MIN}$) as well as the macro-averaged value (e.g. precision$_{MACRO}$). The ultimate purpose of this tool is to identify service-users who have self-harmed perinatally within a cohort. For this reason, we also present positive and negative likelihood ratios (LR$_{POS}$, LR$_{NEG}$) and post-test probabilities.

## Error analysis

**Span errors.** To identify remaining weaknesses in the tool, we performed error analysis on the mention-level evaluation of the test set. The most common recurrent span error was the tool missing mentions of "*suicide*" that had been annotated in the reference standard test set. Whilst death by suicide is not the same as self-harm (which is, by definition, non-fatal), the conceptual line between suicide and self-harm is, in terms of clinician documentation, often blurred. For example, we found clinicians would document: "*no history of suicide*". Clearly, in a clinical entry on a living service-user, a history of death by suicide is impossible. However, this phrase most likely reflects a clinician's attempt to express that the service-user has no history of attempted suicide, i.e. non-fatal self-harm.

**Table 3. Micro-averaged mention-level evaluation results.**

|  | Development set | | | | Test set | | | |
|---|---|---|---|---|---|---|---|---|
|  | Precision | Recall | F-score | Kappa | Precision | Recall | F-score | Kappa |
| **Span** | 0.97 | 0.85 | 0.90 | N/A | 0.94 | 0.81 | 0.87 | N/A |
| **Polarity** | 0.94 | 0.94 | 0.94 | 0.88 | 0.96 | 0.96 | 0.96 | 0.91 |
| **Temporality** | 0.81 | 0.81 | 0.81 | 0.57 | 0.83 | 0.83 | 0.83 | 0.62 |
| **Status** | 0.89 | 0.89 | 0.89 | 0.76 | 0.88 | 0.88 | 0.88 | 0.68 |

**Table 4. Service-user-level evaluation results on the test set (N = 59 service-users).**

| | Manual Coding | Tool | Tool with Heuristic |
|---|---|---|---|
| **Service-users flagged** | 11 | 14 | 4 |
| **Prevalence** | 18.6% | 23.7% | 6.8% |
| **Precision$_{MAJ}$** | N/A | 0.91 | 0.87 |
| **Precision$_{MIN}$** | N/A | 0.50 | 1 |
| **Precision$_{MACRO}$** | N/A | 0.71 | 0.94 |
| **Recall$_{MAJ}$** | N/A | 0.85 | 1 |
| **Recall$_{MIN}$** | N/A | 0.64 | 0.36 |
| **Recall$_{MACRO}$** | N/A | 0.75 | 0.68 |
| **F-score$_{MAJ}$** | N/A | 0.88 | 0.93 |
| **F-score$_{MIN}$** | N/A | 0.56 | 0.53 |
| **F-score$_{MACRO}$** | N/A | 0.72 | 0.73 |
| **Kappa** | N/A | 0.44 | 0.48 |
| **LR$_{POS}$ (95% CI)** | N/A | 4.4 (1.9–9.9) | Infinity |
| **LR$_{NEG}$ (95% CI)** | N/A | 0.4 (0.2–0.9) | 0.6 (0.4–1.0) |
| **Post-test probability$_{POS}$ (95%CI)** | N/A | 50.0 (31–69%) | 100% |
| **Post-test probability$_{NEG}$(95%CI)** | N/A | 8.9 (4–18%) | 12.7 (9–18%) |

There were a small number of instances where the tool erroneously identified phrases not annotated in the test set. This was largely for two reasons. Firstly, there were unusual examples of clinician documentation style that referred to things that were not self-harm e.g. "*OD AM*", to indicate "*once daily in the morning*". We had included "*OD*" in the coding structure, as a synonym for "*overdose*". Secondly, there were some specific and uncommon examples of self-harm that were not included in the coding structure e.g. "*drinking*" specific poisonous substances. Finally, the grammatical context of the verb "*jump*" also proved difficult to capture reliably, as this verb can be used with a variety of prepositions that do not always indicate attempted self-harm e.g. "*jump <u>to</u> kill herself*"/"*jump <u>through</u> a window*" are valid mentions of self-harm, while "*jump <u>down</u> the stairs*" and "*jump <u>to</u> conclusions*" are not.

**Table 5. Service-user-level evaluation results on the development set (N = 152 service-users).**

| | Manual Coding | Tool | Tool with Heuristic |
|---|---|---|---|
| **Service-users flagged** | 29 | 46 | 29 |
| **Prevalence** | 19.1% | 30.3% | 19.1% |
| **Precision$_{MAJ}$** | N/A | 0.97 | 0.93 |
| **Precision$_{MIN}$** | N/A | 0.57 | 0.69 |
| **Precision$_{MACRO}$** | N/A | 0.77 | 0.81 |
| **Recall$_{MAJ}$** | N/A | 0.84 | 0.93 |
| **Recall$_{MIN}$** | N/A | 0.90 | 0.69 |
| **Recall$_{MACRO}$** | N/A | 0.87 | 0.81 |
| **F-score$_{MAJ}$** | N/A | 0.90 | 0.93 |
| **F-score$_{MIN}$** | N/A | 0.69 | 0.69 |
| **F-score$_{MACRO}$** | N/A | 0.80 | 0.81 |
| **Kappa** | N/A | 0.60 | 0.62 |
| **LR$_{POS}$ (95%CI)** | N/A | 5.5 (3.6–8.4) | 9.4 (4.8–19) |
| **LR$_{NEG}$ (95%CI)** | N/A | 0.1 (0.04–0.4) | 0.3 (0.2–0.6) |
| **Post-test probability$_{POS}$ (95%CI)** | N/A | 56.5 (46–66%) | 69.0 (53–82%) |
| **Post-test probability$_{NEG}$(95%CI)** | N/A | 2.8 (1–8%) | 7.3 (4–12%) |

**Attribute errors.** Regarding errors on the status attribute, we assumed the modal auxiliary "*would*" to be a marker of non-relevance in the tool's contextual search, as it would usually indicate a future conditional event that had not yet happened. However, the tool sometimes erroneously considered modals appearing after mentions, for example, "*she thought the [self-harm] would kill her*". A further recurring status error was that we assumed "*risk to self*" headings in EHRs indicated the part of clinical assessment known as "risk assessment", which is a discussion of the service-user's future risk to self and would therefore contain non-relevant mentions. However error analysis revealed this phrase was occasionally used as a section header detailing past self-harm events.

Regarding temporality, our default approach was to mark events as current unless there was a clear historical marker. However, we found that temporality indicators became unclear in cases using a coordination, e.g. "*no current or past suicide attempts*". The tool annotated this mention as current, whilst in the reference standard it was annotated as historical. Assessing temporality was also problematic where there were no contextual markers due to the shorthand note-taking style of the clinician, for example "*2 x OD*".

## Discussion

Self-harm is a conceptually and clinically complex area. Framing it temporally within the narrow time-period of pregnancy and the postnatal year increases the complexity. However, we have shown that it is possible to develop an NLP tool that, with acceptable precision and recall, can identify perinatal self-harm within electronic healthcare records at both a mention and patient level. Given the limitations in existing data on the prevalence of perinatal self-harm [5], this is a significant step forward.

The pair-wise inter-rater agreement suggests that temporality was the hardest attribute for annotators to agree on. This may reflect the high degree of complexity and ambiguity in ways that self-harm is documented.

Micro-averaged mention-level evaluation figures reflect this pattern, although precision, recall and F-score for all attributes were all still >0.8. After adjustment for chance, temporality remains the weakest attribute, although kappa is still almost 0.8.

We felt that the test set (Table 4) was too small to evaluate service-user level performance. Using the much larger development set (Table 5), we showed that, by using a heuristic rule of two, we could generate a tool with macro-averaged F-score 0.81 and a high positive likelihood ratio of 9.4 (95% CI 4.8–19).

Overall, scores for kappa were lower than precision/recall/F-score (patient-level kappa 0.62), suggesting some agreement may have been due to chance. However the limitations of using kappa in dichotomous classification system performance analysis with unbalanced datasets should be noted [38], particularly where the sample size is small [39]. How the tool performs in a much larger sample would be an interesting area of further study.

The use of heuristic rules is commonplace in NLP literature [40–42] and it is well-recognised that in clinical contexts moving from mention to person-level performance often requires "post-processing" [43]. We believe the use of a heuristic in this case does have face validity, as in reality if a service-user has self-harmed perinatally this is a significant clinical event, meaning it is likely to be followed up at subsequent visits or by other clinicians, i.e. further mentions of it would be generated within the service-user's body of EHRs.

We believe this tool could potentially be adapted to ascertain self-harm in other contexts. Work is currently underway to investigate whether it can be adapted to ascertain self-harm in adolescent populations and among women with eating disorders.

### Strengths

We believe this is a novel development in the field of using NLP to investigate self-harm, as it focusses specifically on acts of perinatal self-harm among women with SMI. We used a bespoke NLP strategy developed using both clinical and NLP expertise. Our iterative approach meant that we could use unusual examples encountered in development phases of annotation to refine the tool.

### Limitations

Our corpus was relatively small and generalisability to EHRs from other populations and mental healthcare providers is uncertain. The main outcomes of error analysis are that it is often hard to find reliable contextual markers for ambiguous mentions. The use of syntactic coordination (and, or, etc.) often makes this even more problematic. Temporality is notoriously difficult to analyse with NLP and is a field of research in its own right [44, 45]. The analysis also reveals something about how clinical note-taking is done e.g. the high variability in the words and formulations used by clinicians.

## Conclusions

We have shown, using novel methods and a combination of clinical and linguistic processing expertise, that is possible to develop an NLP tool that will, with acceptable precision and recall, identify perinatal self-harm in electronic healthcare records, albeit with limitations, particularly in terms of defining temporality.

## Supporting information

**S1 Table. Number of true mentions of self-harm per service-user, within the reference standard dataset.**
(DOCX)

**S2 Table. Macro-averaged pairwise inter-annotator agreement.**
(DOCX)

**S3 Table. Macro-averaged mention-level performance.**
(DOCX)

**S1 File. List of synonyms for self-harm.**
(DOCX)

**S2 File. Annotation guidelines.**
(DOCX)

## Author Contributions

**Conceptualization:** Louise M. Howard.

**Data curation:** Karyn Ayre, André Bittar.

**Formal analysis:** Karyn Ayre, André Bittar, Joyce Kam, Somain Verma.

**Funding acquisition:** Karyn Ayre.

**Investigation:** Karyn Ayre, André Bittar.

**Methodology:** André Bittar, Rina Dutta.

**Project administration:** Karyn Ayre, André Bittar.

**Resources:** André Bittar.

**Software:** Karyn Ayre, André Bittar.

**Supervision:** Louise M. Howard, Rina Dutta.

**Validation:** Karyn Ayre.

**Visualization:** Karyn Ayre.

**Writing – original draft:** Karyn Ayre.

**Writing – review & editing:** Karyn Ayre, André Bittar, Louise M. Howard, Rina Dutta.

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
