## [Decision Letter · Decision Letter 0]

18 May 2021

PONE-D-21-06370

Developing a Natural Language Processing tool to identify perinatal self-harm in electronic healthcare records.

PLOS ONE

Dear Dr. Ayre,

Thank you for submitting your manuscript to PLOS ONE. After careful consideration, we feel that it has merit but does not fully meet PLOS ONE’s publication criteria as it currently stands. Therefore, we invite you to submit a revised version of the manuscript that addresses the points raised during the review process.

We look forward to receiving your revised manuscript.

Kind regards,

Natalia Grabar

Academic Editor

PLOS ONE

Journal Requirements:

Additional Editor Comments:

Dear Authors,

Please consider the reviewers' comments and take them into account when preparing the new version of the submission.

The issues on generalizability and reusability of the methods should be addressed.

Please, prepare the letter with answers.

Reviewers' comments:

Reviewer's Responses to Questions

**Comments to the Author**

1. Is the manuscript technically sound, and do the data support the conclusions?

Reviewer #1: Yes

Reviewer #2: Yes

2. Has the statistical analysis been performed appropriately and rigorously? 

Reviewer #1: N/A

Reviewer #2: Yes

3. Have the authors made all data underlying the findings in their manuscript fully available?

Reviewer #1: No

Reviewer #2: No

4. Is the manuscript presented in an intelligible fashion and written in standard English?

Reviewer #1: Yes

Reviewer #2: Yes

5. Review Comments to the Author

Reviewer #1: [REVIEW] Developing a Natural Language Processing tool to identify perinatal self-harm in electronic healthcare records.

PLOS ONE

21st April 2021

# SUMMARY

This paper presents work on the development of an NLP algorithm to detect individuals who have experienced perinatal self-harm. 400 clinical notes were sampled from the South London & Maudsley hospital system secondary mental healthcare service representing 232 distinct patients. Interannotator agreement ranged from 0.78 to 0.92. The notes were divided into development and test sets (320 and 80, respectively) and a rule-based system built around the Spacy Python NLP library was developed. The mention level performance of the system performed quite well, as did the patient-level classification, resulting in an F-score of 0.81.

This was a clear, well-presented paper on an interesting and important topic (i.e. the identification of perinatal self-harm). The methodology adopted was appropriate and reasonable for the stated research goals. I have made some additional comments/suggestions below. Thanks for the opportunity to review this work.

# COMMENTS

* ln 56. This isn’t a major issue, but I believe “reference standard” is the preferred term (rather than “gold standard”)

* ln 57. “As service users usually had more than one EHR…” This is a bit ambiguous. It could mean either EHR System (i.e. a patient uses two medical systems with different EHR systems) or particular EHR encounters. I believe — based on usage later in the paper — that you mean it in the latter sense.

* ln 61. The results section of the abstract is a little schematic

* ln 129. “We devised rules around the span of text to annotate as a mention and the attributes to annotate the mention with.” I think there is an grammar issue here

* ln 145. “For example, if a mention described someone who had thoughts of self-harm, by definition, no act of self-harm had occurred” I’m not so sure about this. I think “by definition” may be too strong given that it is quite possible that someone may have suicidal ideation and also harm themselves. I’d suggest weakening this (e.g. “indicating…”)

* ln 169. “First, we randomly sampled 400 EHRs from Taylor’s study of self

harm in pregnant SLaM service-users with affective and non-affective psychotic disorders (23).” See comment re EHRs above

* I may have missed this, but do you have the annotation guidelines as supplementary materials

* ln 187. I’m not sure “tokens” should be in quotation marks

* ln 195. “A full list of these lexicons and example content is shown in Table 1”

* ln 209. You might include a couple of sentences on dependency parsing here (what it is, how it has been used with EHR data in the past)

* ln 216. Isn’t negation covered in the previous section

* ln 308. The structure of the discussion section is a bit confusing (2 “discussion” headers)

* ln 331. I agree with the strengths listed, but limitations should also — I believe — include the relatively small annotated corpus and unknown generalisability to other (non-SLAM) records.

* ln 347. Some of the references are incomplete

* It would be nice to have more discussion of (potential) downstream applications of the tool in the discussion section

Reviewer #2: The article proposes to implement an NLP tool apply in a very specific context to identify perinatal self-harm.

From a general point of view, the article is sound, easy to read and precise.

We do not have access to the all data for ethical concerns.

In the end, the results are in general good. The authors choose to work on different tasks and some of them are complex. Thus it is not a surprise to have some disagreement on Kappa scores or other.

The final tool seems to be useful in the very specific situation of the authors.

My main concerns is about the re-usability of the result. Authors pre-process the data and write ad hoc rules for the data. I am not sure how it generalizes over a given medicine process or on the description of the disease itself. It is quite important in my understanding of the paper because, although the outcome is relevant in the given situation, it seems to rely on artefacts of the institution rather than the description of the disease. This does not call into question the intrinsic value of the result for the hospital service, but it would give it more perspective.

Without talking about the reproducibility of the research, it would be interesting to work on the possibility of reusing the proposed strategy for another pathology, especially since the authors are working on a language with many resources.

On a more specific point, I am not exactly sure to understand how the negation is used in the tool. This is a crucial point for a correct interpretation, exactly as for the coordination. The authors address this issue but it is not clear how it is done.

6. PLOS authors have the option to publish the peer review history of their article (what does this mean?). If published, this will include your full peer review and any attached files.

Reviewer #1: No

Reviewer #2: No

---

## [Author Response · Author response to Decision Letter 0]

2 Jun 2021

Thank you for your letter of 18.05.21 and the two reviewers’ immensely helpful comments. We have undertaken all the revisions to the manuscript (ref PONE-D-21-06370) recommended by them to improve the quality and readability of our manuscript.

Below I shall incorporate both reviewers’ comments in order (C1, C2 etc), state our responses (R1, R2 etc) and reference the changes we have made in the revised manuscript. Line numbers refer to the revised manuscript with tracked changes.

Reviewer #1: 

C1. ln 56. This isn’t a major issue, but I believe “reference standard” is the preferred term (rather than “gold standard”)

R1. We have changed “gold standard” to “reference standard” throughout the manuscript and in the title of Supporting Information Table 1.

C2. ln 57. “As service users usually had more than one EHR…” This is a bit ambiguous. It could mean either EHR System (i.e. a patient uses two medical systems with different EHR systems) or particular EHR encounters. I believe — based on usage later in the paper — that you mean it in the latter sense.

R2. Thank you, we appreciate this was not ideally phrased and could lead to confusion. We have now removed the phrase “As service users usually had more than one EHR…” from the relevant sentence in the abstract (lines 61-62). We have also included a sentence in the “Data Sources” paragraph of the Methods section, stating: “In this context, “EHR” refers to a single clinical document, within one universal electronic healthcare recording system called the “Electronic Patient Journey System” (lines 113-115).

C3. ln 61. The results section of the abstract is a little schematic

R3. We have included a sentence at the end of the results section of the abstract, to improve the narrative quality: “Considering the task difficulty, the tool performs well, although temporality was the attribute with the lowest level of annotator agreement” (lines 68-9).

C4. ln 129. “We devised rules around the span of text to annotate as a mention and the attributes to annotate the mention with.” I think there is an grammar issue here

R4. We have changed this to: “We devised rules regarding the span of text to annotate as a mention and how to annotate mention attributes” (lines 139-40).

C5. ln 145. “For example, if a mention described someone who had thoughts of self-harm, by definition, no act of self-harm had occurred” I’m not so sure about this. I think “by definition” may be too strong given that it is quite possible that someone may have suicidal ideation and also harm themselves. I’d suggest weakening this (e.g. “indicating…”)

R5. We entirely agree and have changed the phrasing to: “For example, if a mention described thoughts of self-harm, rather than an act of self-harm, they were inferred and annotated to be non-relevant” (lines 157-9).

C6. ln 169. “First, we randomly sampled 400 EHRs from Taylor’s study of self-harm in pregnant SLaM service-users with affective and non-affective psychotic disorders (23).” See comment re EHRs above

R6. We believe we have addressed this issue in our response to Point 2.

C7. I may have missed this, but do you have the annotation guidelines as supplementary materials

R7. We did not previously, but we have now included the annotation guidelines as “Supporting Information 2”, and have re-numbered the other Supporting Information and Supporting Information Tables in the manuscript accordingly

C8. ln 187. I’m not sure “tokens” should be in quotation marks

R8. We have removed the quotation marks

C9. ln 195. “A full list of these lexicons and example content is shown in Table 1”

R9. Apologies but we are not sure what Reviewer 1 is referring to in point 9. We think it is possible that their comment is missing?

C10. ln 209. You might include a couple of sentences on dependency parsing here (what it is, how it has been used with EHR data in the past)

R10. At lines 206-9, we have added an explanatory sentence and given some examples of previous work which uses dependency parsing on clinical texts.

C11. ln 216. Isn’t negation covered in the previous section

R11. We agree and have replaced “To further assign values to attributes for identified mentions, a contextual search is used to detect markers of negation, temporality and status” with: “To further assign values to attributes for identified mentions, a contextual search is used to detect markers of temporality and status” (lines 236-7).

C12. ln 308. The structure of the discussion section is a bit confusing (2 “discussion” headers)

R12. We have removed the “main findings” and “discussion” sub-headings, merging the content of the discussion in a logical structure, with strengths and limitations having separate sub-headings. We hope this makes it easier to follow.

C13. ln 331. I agree with the strengths listed, but limitations should also — I believe — include the relatively small annotated corpus and unknown generalisability to other (non-SLAM) records.

R13. We have included the following statement in the limitations section: “Our corpus was relatively small and generalisability to EHRs from other populations and mental healthcare providers is uncertain” (lines 364-5).

C14. ln 347. Some of the references are incomplete

R14. Thank you for highlighting this. We have gone through the references and believe all are now complete.

C15. It would be nice to have more discussion of (potential) downstream applications of the tool in the discussion section

R15. We have included the following statement: “We believe this tool could potentially be adapted to ascertain self-harm in other contexts. Work is currently underway to investigate whether it can be adapted to ascertain self-harm in adolescent populations and among women with eating disorders” (line 354-6).

Reviewer #2: 

C1. The final tool seems to be useful in the very specific situation of the authors. My main concerns is about the re-usability of the result. Authors pre-process the data and write ad hoc rules for the data. I am not sure how it generalizes over a given medicine process or on the description of the disease itself. It is quite important in my understanding of the paper because, although the outcome is relevant in the given situation, it seems to rely on artefacts of the institution rather than the description of the disease. This does not call into question the intrinsic value of the result for the hospital service, but it would give it more perspective.

R1. Thank you for this feedback. We believe this point is similar to point 13 made by reviewer 1, to which we responded by including the following sentence in the limitations section: “Our corpus was relatively small and generalisability to EHRs from other populations and mental healthcare providers is uncertain” (lines 364-5).

C2. Without talking about the reproducibility of the research, it would be interesting to work on the possibility of reusing the proposed strategy for another pathology, especially since the authors are working on a language with many resources.

R2. We believe this point is similar to point 15 made by Reviewer 1, to which we responded by including the following sentences in the Discussion section: “We believe this tool could potentially be adapted to ascertain self-harm in other contexts. Work is currently underway to investigate whether it can be adapted to ascertain self-harm in adolescent populations and among service-users with eating disorders” (lines 354-6). 

C3. On a more specific point, I am not exactly sure to understand how the negation is used in the tool. This is a crucial point for a correct interpretation, exactly as for the coordination. The authors address this issue but it is not clear how it is done.

R3. Thank you. We have included a further citation of the use of dependency parsing in the context negation detection (lines 206-9).

---

## [Editor Report · Decision Letter 1]

14 Jun 2021

Developing a Natural Language Processing tool to identify perinatal self-harm in electronic healthcare records.

PONE-D-21-06370R1

Dear Dr. Ayre,

We’re pleased to inform you that your manuscript has been judged scientifically suitable for publication and will be formally accepted for publication once it meets all outstanding technical requirements.

Kind regards,

Natalia Grabar

Academic Editor

PLOS ONE

---

## [Editor Report · Acceptance letter]

30 Jun 2021

PONE-D-21-06370R1 

Developing a Natural Language Processing tool to identify perinatal self-harm in electronic healthcare records. 

Dear Dr. Ayre:

I'm pleased to inform you that your manuscript has been deemed suitable for publication in PLOS ONE. Congratulations! Your manuscript is now with our production department. 

Kind regards, 

on behalf of

Dr. Natalia Grabar 

Academic Editor

PLOS ONE